# Bacterioneuston in Lake Baikal: Abundance, Spatial and Temporal Distribution

**DOI:** 10.3390/ijerph15112587

**Published:** 2018-11-19

**Authors:** Agnia D. Galachyants, Irina V. Tomberg, Elena V. Sukhanova, Yulia R. Shtykova, Maria Yu. Suslova, Ekaterina A. Zimens, Vadim V. Blinov, Maria V. Sakirko, Valentina M. Domysheva, Olga I. Belykh

**Affiliations:** Limnological Institute Siberian Branch of the Russian Academy of Sciences, 3, Ulan-Batorskaya, Irkutsk 664033, Russia; kaktus@lin.irk.ru (I.V.T.); sukhanova@lin.irk.ru (E.V.S.); julis83@yandex.ru (Y.R.S.); msuslova1979@mail.ru (M.Y.S.); ekaterinasiemens93@gmail.com (E.A.Z.); bwad@lin.irk.ru (V.V.B.); sakira@lin.irk.ru (M.V.S.); hydrochem@lin.irk.ru (V.M.D.)

**Keywords:** environmental factors, bacterioneuston, Lake Baikal, surface microlayer, total bacterial abundance, number of cultured heterotrophic bacteria

## Abstract

An aquatic surface microlayer covers more than 70% of the world’s surface. Our knowledge about the biology of the surface microlayer of Lake Baikal, the most ancient lake on Earth with a surface area of 31,500 km^2^, is still scarce. The total bacterial abundance, the number of cultured heterotrophic temporal bacteria, and the spatial distribution of bacteria in the surface microlayer and underlying waters of Lake Baikal were studied. For the first time, the chemical composition of the surface microlayer of Lake Baikal was determined. There were significant differences and a direct relationship between the total bacterial abundance in the surface microlayer and underlying waters of Lake Baikal, as well as between the number of cultured heterotrophic bacteria in studied water layers in the period of summer stratification. In the surface microlayer, the share of cultured heterotrophic bacteria was higher than in the underlying waters. The surface microlayer was characterized by enrichment with PO_4_^3−^, total organic carbon and suspended particulate matter compared to underlying waters. A direct relationship was found between the number of bacteria in the surface microlayer and environmental factors, including temperature, total organic carbon and suspended particulate matter concentration.

## 1. Introduction

The aquatic surface microlayer (SML) is a physical boundary between the hydrosphere and the atmosphere. Thickness of the SML is several tens of micrometers; however, it differs significantly from the water column in physicochemical characteristics and contains high concentrations of organic compounds, such as lipids, proteins and polysaccharides [1]. The SML occupies approximately 71% of earth’s surface, covering all marine and fresh water bodies. Due to its specific location, the climatic and weather phenomena, such as wind, precipitation, and temperature changes, strongly affect the SML [2]. Due to its unique physicochemical characteristics, the SML forms a distinct biotope inhabited with a specific microbial community called neuston. Bacteria inhabiting the SML, or bacterioneuston, are an important component of water ecosystems [3]. They play a considerable role in maintaining SML physicochemical properties and are actively involved in the exchange of substances and gases between the atmosphere and the hydrosphere [2]. Bacterioneuston communities are essential to the global carbon cycle [4,5].

Estimation of the total bacterial abundance in natural water bodies currently remains important, being an integral characteristic of microbial communities. Epifluorescence microscopy is a widely used method for direct enumeration of total bacteria [6]. The total bacterial abundance in the SML was studied in marine [7,8,9,10] and fresh water bodies [11,12,13]. No data were found on the number of cultured heterotrophic bacteria in the SML of fresh water bodies, unlike marine ecosystems [14,15,16]. Many studies have shown that the number of bacteria in the SML is often greater than in the underlying waters (UW) [8,10,11,13].

Bacterioplankton abundance can be influenced by environmental factors, such as water temperature and organic matter content [17,18,19,20]. In the SML of Alpine lakes, a close relationship was found between water temperature and the number of Betaproteobacteria [12]. There was also a positive correlation between the number of Actinobacteria and the organic nitrogen concentration in the SML. In the high-altitude lakes of the Pyrenees, there was a correlation between surface water temperature and biological parameters, such as the number and activity of bacteria [13].

Lake Baikal is the deepest lake in the world at 1637 m (5371 ft) with a surface area of 31,500 km^2^ (12,162 sq.mi.). Lake Baikal is the largest freshwater lake by volume in the world, containing 20% of the world’s fresh surface water. Baikal’s age is estimated at 25 million years, making it the most ancient lake in geological history [21]. The lake was declared a UNESCO World Heritage Site in 1996. The spring period here differs markedly to summer in terms of water temperature and hydrological regime. Biological spring (late May to early June) is characterized by a period of homothermy when, due to wind mixing and movement of water mass, the temperature throughout the water column is the same and is about 4 °C. In the biological summer (from end of June to second half of August), a period of thermal stratification is observed when the surface water warms up and its temperature reaches a maximum of 12–24 °C [22]. The chemical composition of Lake Baikal water is characterized by low nutrient and ionic concentrations (around 96 mg/L). The content of NO_3_^−^ and PO_4_^3−^ in the 0–50 m water layer does not exceed 0.03 and 0.30 mg/L, respectively. In summer, the development of phytoplankton significantly reduces the concentrations of these components in the euphotic layer. NH_4_^+^ and NO_2_^−^ in low concentrations are registered in the upper water layer when plankton is dying off [21,23]. These features make Baikal unique compared to other water bodies where the neuston has been studied. Microbiological investigations of the lake began in 1927 [24]. Since that time, the number and biomass of bacteria in bottom sediments and in the water column throughout the entire water area of the lake, its interannual dynamics and vertical distribution, were identified [25,26,27]. Last year a lot of works were dedicated to the investigation of the taxonomic composition of the microbiomes in Lake Baikal: biofilm bacterial communities and bacterioplancton [28,29].

Baikal bacterioneuston was first studied in the 1970s [30]. In particular, Nikitin estimated the total bacterial abundance and number of cultured heterotrophic bacteria at several stations in the southern basin of the lake [30]. Later in 2016 the investigation of neuston in Lake Baikal was continued: diversity and physiological and biochemical properties of heterotrophic bacteria isolated from neuston have been studied and then in 2017 taxonomic composition of bacterioneuston communities was examined [31,32]. Nevertheless, there have been no studies of the bacterioneuston abundance in Lake Baikal since the 1970s.

The aim of the present study was to analyze the spatial distribution of Lake Baikal bacterioneuston in different seasons and years, and to investigate the relationship between bacterioneuston abundance and environmental factors.

## 2. Materials and Methods 

Sampling of the SML was carried out throughout Lake Baikal in the May-June period of 2013 to 2016 and in August of 2013, 2015 and 2016 (Figure 1, Appendix A). The number of cultured heterotrophic bacteria was determined in the SML and UW in the May–June period and in August of each year. Enumeration of total bacteria in the SML and UW was performed in May–June and August of 2013, in May–June of 2014, and in May–June and August of 2015.

SML sample collection took place from a boat, mainly during calm weather, using a metal mesh screen [33]. To obtain the integral water sample (−500 mL), sampling was performed continuously for 20–30 min. The thickness of the water layer was determined as the ratio of sample volume to mesh screen area [34]. The thickness of the water layer sampled in Lake Baikal using the mesh screen was 362–420 μm. The UW samples were collected by syringe from a depth of 15–20 cm.

To estimate cultured heterotrophic bacteria number, 1 mL of SML sample diluted 10- or 100-fold, and 1 mL of UW sample were pour plated on three replicate plates on R-2A medium (Sigma-Aldrich, Saint Louis, MO, USA) (pH 7.2). Samples were processed within 4 h after sampling. The number of colony forming units (CFU) was counted on the 5th to 7th day of culturing in the dark at room temperature. 

Quantification of total bacteria was performed by direct counting on membrane filters using epifluorescence microscopy. Samples were fixed with formalin or glutaraldehyde (final concentration 2% *v*/*v*) and stored away from light. The preserved subsamples of 1 ml were filtered through 0.22-mm pore-size polycarbonate filters (Millipore). The filters for bacteria counting were stained with 4’,6-diamidino-2-phenylindole, dihydrochloride (DAPI) solution [6]. Filters were air-dried and placed on a drop of non-fluorescing immersion oil. Ready preparations were stored in darkness at 4 °C. Filters were examined using Axio Imager M1 microscope (Carl Zeiss, Germany), equipped with a HBO 100W mercury lamp and AxioCam MRm and MRc5 cameras. Registration was done by means of oil lens × 100 (Plan-Fluar). UV excitation was used for DAPI-stained bacteria. Cells were counted from 30 view fields with subsequent averaging. The total number of the cells registered on a filter was 1500 or more.

Wind speed was measured using the Kestrel 4000 Pocket Weather Tracker. Surface water temperature was measured using a handheld ETP-104 temperature sensor.

Chemical analysis of SML and UW samples from a depth of 15–20 cm was conducted in August 2013, in May–June and August 2015, as well as in May–June 2016. PO_4_^3−^, NO_3_^−^, NO_2_^−^ and NH_4_^+^ were analyzed in water samples filtered through mixed cellulose ester membrane filters (Advantec, Tokyo, Japan, pore diameter 0.45 μm). The pH value and content of total organic carbon (TOC) and suspended particulate matter were determined in unfiltered water. A pH meter with a combined electrode and a temperature compensator was used to measure pH (Expert-pH, Russia). Content of suspended particulate matter and the concentration of dissolved nutrients were determined using a photoelectric colorimeter (KFK-3-01-ZOM3, Zagorskii optiko-mekhanicheskii zavod [Zagorskii optical mechanics factory], Sergiev Posad, Russia) according to Wetzel and Likens [35]. Phosphates were identified by the Denigès–Atkins method in modification with tin chloride. Ammonium ions were detected by the indophenol method [36]. Nitrites were determined with a Griss reagent. NO_3_^−^ and NO_2_ content was measured by high performance liquid chromatography (EcoNova, Russia) with UV detection on an inverse-phase column modified with octadecyltrimethylammonium bromide [37]. Total organic carbon was determined by total carbon/nitrogen analyzer (Vario TOC cube, Elementar, Langenselbold, Germany).

Statistical analysis of the data was carried out using the program R-Studio 3.3.1 (https://cran.r-project.org/bin/windows/base/old/3.1.1/). The Pearson correlation coefficient was used to determine the relationship between the characteristics. Significant differences between the samples were identified using the Mann–Whitney–Wilcoxon test. The significance level was 0.05. 

## 3. Results

### 3.1. Abundance and Spatial Distribution of Bacteria in SML and UW of Lake Baikal

The average values of total bacterial abundance in the SML varied in different years with a range of 0.93–1.49 × 10^6^ cells/mL in May–June and 1.73–2.24 × 10^6^ cells/mL in August; in UW, at a depth of 15–20 cm, there were 0.79–0.89 × 10^6^ cells/mL in May–June and 1.15–1.4 × 10^6^ cells/mL in August (Figure 2).

The average values of the cultured heterotrophic bacteria number varied in different years in distinct basins in May-June, with a range of 25–4726 CFU/mL in the SML and 1–295 CFU/mL in UW; in August, 2394–99,226 CFU /mL were recorded in the SML and 161–15,813 CFU/mL in UW (Figure 3).

There were no statistically significant dissimilarities in the number of cultured heterotrophic bacteria and the total bacterial abundance in the SML in different years, as well as between distinct Baikal basins.

Differences in the number of cultured heterotrophic bacteria between the SML and UW were significant (by several times) in all seasons. We identified statistically significant differences in the total bacterial abundance between the SML and UW only in the period of summer stratification.

In August of 2013, 2015 and 2016, we observed a direct correlation between the number of cultured heterotrophic bacteria in the SML and UW; the Pearson correlation coefficient was 0.77 in 2013, 0.89 in 2015 and 0.97 in 2016 (*p* < 0.05). In the spring homothermy periods, there was no correlation. We detected a direct correlation between the total bacterial abundance in SML and UW (the Pearson correlation coefficient was 0.67 in August of 2013, and 0.9 in August of 2015; *p* < 0.05).

### 3.2. Influence of Physical Factors on the Number of Bacteria in SML of Lake Baikal

#### 3.2.1. Wind

When sampling using the metal mesh screen with a wind force of less than 4.9 m/s (average 1.43 ± 0.65 m/s), we did not observe a statistically significant correlation between the number of cultured heterotrophic bacteria in the SML and wind force, neither in the spring or summer seasons. There was also no significant relationship between the total bacterial abundance in the SML and wind force.

#### 3.2.2. Temperature

In the spring seasons, the average temperature of the surface water was in the range 2.4–3 °С (Figure 4). In the summer seasons of 2013, 2015, and 2016, the average temperature of surface water did not vary significantly and was within the range 15–15.7 °С (Figure 4).

The temperature of surface water in May–June and August was almost the same in the southern, northern, and central basins of Lake Baikal. There was a direct correlation between the temperature of surface water and the number of cultured heterotrophic bacteria in the SML (Pearson correlation coefficient 0.57; *p* < 0.05) (Figure 4). There was also a direct correlation between the total abundance of SML bacteria and the surface water temperature, but weaker than in the case of the number of cultured heterotrophic bacteria (Pearson correlation coefficient 0.5; *p* < 0.05).

### 3.3. Chemical Composition of the SML in Lake Baikal

Enrichment with some chemical compounds in the SML compared to UW was detected in all the seasons studied, especially in summer. Thus, in August of 2013, concentrations of suspended particulate matter, PO_4_^3−^ and NO_2_^−^ in the SML were two to four times higher than in UW (Table 1). The pH values in SML were on average 0.2–0.3 units lower than in UW.

In May–June of 2015, statistically significant differences between the PO_4_^3−^ content in the investigated layers were recorded; the concentrations in the SML were on average two to three times higher than in UW. Additionally, high concentrations of NH_4_^+^ were observed in the SML (Table 1). In August of 2015, the TOC concentration in the SML was two to three times higher than in UW. Also, the significant differences in the PO_4_^3−^ content between the SML and UW remained. 

In May–June of 2016, concentrations of PO_4_^3−^ and NO_3_^−^ were still higher in the SML compared to UW, but the gradients were lower than in previous years (Table 1).

Statistical analysis showed a significant and direct correlation between both the total bacterial abundance and the number of cultured heterotrophic bacteria in the SML and the concentration of suspended particulate matter (Pearson correlation coefficient 0.69 for cultured bacteria and 0.87 for the total bacterial abundance; *р* < 0.05). In addition, we found a significant direct correlation between the number of cultured heterotrophic bacteria and the content of TOC in the SML and UW (*р* = 0.02; Pearson correlation coefficient 0.7).

## 4. Discussion

Values of the total bacterial abundance in the SML of Lake Baikal obtained in the present study were three orders of magnitude lower than those reported previously in the 1980s. According to the data of [30], the total bacterial abundance in the SML of Lake Baikal was in the range 570–2238 × 10^6^ cells/mL. There are no reports of such a high total bacterial abundance in the SML in either marine or in fresh water bodies. For the total bacterial abundance estimation, Nikitin used the method of direct counting of bacteria collected on membrane filters and stained with erythrosin [38]. It was shown earlier that the erythrosin staining method is comparable with the DAPI staining method for quantifying total bacteria [26]. Therefore, differences in the methodology used cannot explain the differences in the values obtained. Probably, the incorrect results were caused by some technical errors in sampling or sample processing. In the UW (0–2 cm thick), the total bacterial abundance was in the range of 0.7–2.8 × 10^6^ cells/mL [30], which is comparable to our data. According to data from recent years, the total bacterial abundance in the surface layer (0–50 m) of Lake Baikal was 0.2–2.2 × 10^6^ cells/mL [39,40], which also corresponds to our data.

Moreover, our data corresponds to those for other bodies of fresh water. In the high altitude lakes of the Central Pyrenees, the total bacterial abundance in the SML varied in the range 0.2–12.2 × 10^6^ cells/mL, and in the high mountain lakes of the Alps 0.2–3.2 × 10^6^ cells/mL [11,12,13]. In UW, the same authors reported 0.1–1.8 × 10^6^ cells/mL and 0.2–0.7 × 10^6^ cells/mL, respectively.

The number of cultured heterotrophic bacteria in the SML of Lake Baikal, according to [30] was in the range 14,200–286,000 CFU/mL, which is comparable with our data for the summer seasons. In marine ecosystems, the number of cultured bacteria in the SML is higher than in Lake Baikal. For instance, in the Pacific Ocean off Vancouver Island, the number of cultured bacteria in the SML varied from 2.1 × 10^3^ to 2.3 × 10^6^ CFU/mL [14]; in the SML of the estuary in North Carolina it reached 2.0–5.6 × 10^6^ CFU/mL [14]. In the Mediterranean Sea, the number of cultured bacteria in the SML ranged from 3.3 to 611 × 10^3^ CFU/mL [16].

Our data imply that the differences in the number of bacteria between the SML and UW are more significant for the cultured forms than for total bacterial abundance. A study of the bacterioneuston from the Mediterranean Sea showed that the ratio of cultured bacteria number in the SML and UW was much higher than the ratio of the total number of bacteria in the same water layers. For the cultured forms, this ratio was 11.07, while for the total number of bacteria it was 1.09 [34]. In fact, other authors, who used the cultivation method, also reported high numbers of bacteria in the SML, an order or several orders of magnitude greater than in UW [14,15,16,30]. At the same time, the total bacterial abundance did not show such significant differences between the SML and UW [5,7,8,9,10,11,12,13,16,41,42], although many authors indicated a higher total bacterial abundance in the SML. Probably, media used for cultivation that were rich in organic matter and nutrients were more appropriate for the SML bacteria which are adapted to live in an abundance of nutrients. Perhaps, effective cultivation of bacteria from the water column requires a more careful selection of media.

The data obtained by other authors confirm a direct relationship between the number of bacteria in the SML and UW [9,16]. This relationship provides some evidence that the bacterioneuston originates from the water column. Bacteria enter the SML by a passive transport, together with buoyant particles or gas bubbles [8,16]. Absence of a reliable correlation between the number of cultured heterotrophic bacteria in the SML and UW in May–June, as well as no differences in the total bacterial abundance between the SML and UW in May–June, can be explained by the intensive mixing of the surface waters during the transition from an inverse thermal stratification to a direct one [43].

Among the various physical factors affecting the SML, the wind is a particular one since it causes the mixing of the underlying water layers, resulting in the dilution of the SML. However, many studies indicate the stability of the SML with wind forces of up to 6–10 m/s, which corresponds to our data [7,41,44,45]. The authors sampled the SML using a glass plate [41,44,45] or a mesh screen [7,44], collecting the water layer of approximately 100 and 300 μm, respectively, which exceeds the true thickness of the SML. Nevertheless, in one of the studies, in which the samples were collected with a glass plate, the authors found a direct relationship between the number of SML bacteria and wind force [9]. They carried out sampling at a wind speed of 1.0–40.3 m/s (average wind speed was 13.1 ± 12.44 m/s). It is probable that we did not detect this relationship due to the low wind force at all sampling stations. 

Another important environmental factor is the surface water temperature, since this has an obvious seasonal character and, hence, drastically alters the living conditions of the inhabitants, including bacteria. The data obtained indicate that more suitable temperatures for the SML bacteria of Lake Baikal are those typical for August, rather than for May–June. The data obtained by other researchers also confirm the correlation of biological parameters (the number of bacteria) in the SML primarily with the surface water temperature [12,13].

The SML differs from the water column by containing higher concentrations of nutrients and organic matter [1,7,12,46]. For example, in the open water of the north-western part of the Black Sea, the content of PO_4_^3−^ in the SML was on average two to four times higher than at a depth of approximately 0.5 m, and in the North Atlantic chemical oxygen demand in the upper 0.15 mm of water was on average an order of magnitude higher than at a depth of 0.5 m [46]. For the first time, we studied the chemical composition of the SML of Lake Baikal and found differences between the SML and UW, which corresponded to other authors’ reports. 

A correlation between the number of bacteria in the SML and the concentration of suspended particulate matter can be explained by available data. Turbidity, or concentration of suspended particulate matter, is the cloudiness or haziness of a fluid caused by large numbers of individual particles. Many researchers have shown that the proportion of bacteria attached to particles in the SML was 3 to 10 times higher than in the UW [8,9]. Since the bacteria attached to the particles represent a significant part of all the microorganisms inhabiting the surface microlayer (23 ± 1% according to [8]; 42–29% according to [9], thus the correlation between the concentration of particulate matter and the number of bacteria in the SML seems to be logical. Interestingly, the communities of bacteria attached to the particles in SML and UW have more taxonomic differences than the communities of free-living bacteria [42]. It is not surprising, since particles play an important role in the SML, acting as substrates for the attached forms of bacteria.

Previously, we showed the predominance of copiotrophic bacteria in the cultured bacterial community of SML [31]. This can explain a relationship between the content of TOC and the number of cultured heterotrophic bacteria in the SML of Lake Baikal.

## 5. Conclusions

To conclude, we have identified significant differences and a direct relationship between the total bacterial abundance in the SML and UW, as well as between the number of cultured heterotrophic bacteria in these water layers during the summer season. In the SML, the proportion of cultured heterotrophic bacteria was higher than in UW. The water temperature determined by the season significantly affects the total bacterial abundance, and especially the number of cultured heterotrophic bacteria in SML and UW. We detected the differences in the chemical composition between SML and UW of Lake Baikal. In SML, there were higher concentrations of PO_4_^3−^ and TOC, as well as suspended particulate matter, compared to the UW. We determined a direct relationship between the number of bacteria in the SML and suspended particulate matter concentration, as well as between the number of cultured heterotrophic bacteria and the content of TOC in the SML.

## Figures and Tables

**Figure 1 ijerph-15-02587-f001:**
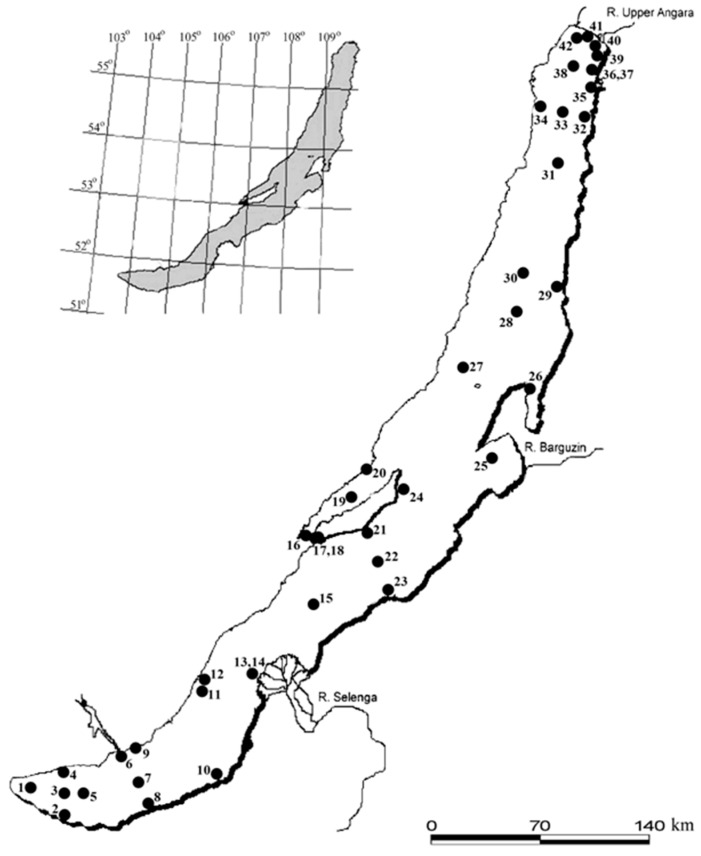
Location of the aquatic surface microlayer (SML) and underlying waters (UW) sampling stations in Lake Baikal in 2013–2016. Note. 1—12 km from Kultuk village; 2—3 km from Solzan village; 3—central station of section Marituy village–Solzan village; 4—3 km from Maritui village; 5—central station of section Ivanovskii Cape−Murino village; 6—Listvenichnyi Bay; 7—central station of section Listvyanka village–Tankhoi village; 8—3 km from Tankhoi village; 9—opposite Bolshie Koty village; 10—opposite Babyshkin village; 11—Peschanaya Bay; 12—Babushka Bay; 13—1 km from Kharauz branch (Selenga River); 14—3 km from Kharauz branch (Selenga River); 15—central station of section the Anga River–Sukhaya River; 16—Mukhor Bay (Maloe More strait); 17—Bazarnaya Bay (Olkhonskie Vorota strait); 18—central station of Olkhonskie Vorota strait; 19—central station of Maloe More strait; 20—opposite Zunduk Cape (Maloe More strait); 21—3 km from the Ukhan Cape; 22—central station of section the Ukhan Cape–Tonkii Cape; 23—3 km from the Tonkii Cape; 24—Shunte Pravyi Cape; 25—Barguzinskii Bay; 26—Chivyrkuiskii Bay; 27—central station of section the Pokoiniki Cape–Great Ushkanii Island; 28—central station of section the Cape Zavorotnyi−Sosnovka River; 29—3 km from Davsha village; 30—central station of section the Cape Elokhin−Davsha village; 31—central station of section the Kotelnikovskii Cape−Amnundakan River; 32—3 km from the Turali Cape; 33—central station of section Baikalskoe village−Turali Cape; 34—3 km from Baikalskoe village; 35—Ayaya Bay; 36—Frolikha Bay; 37—opposite Frolikha Bay; 38—central station of section Tyya River–Nemnyanka Cape; 39—Birakan Cape; 40—5 km from the Verkhnyaya Angara River; 41—Angara-Kicher shoal (Millionnyi island); 42—3 km from Nizhneangarsk city.

**Figure 2 ijerph-15-02587-f002:**
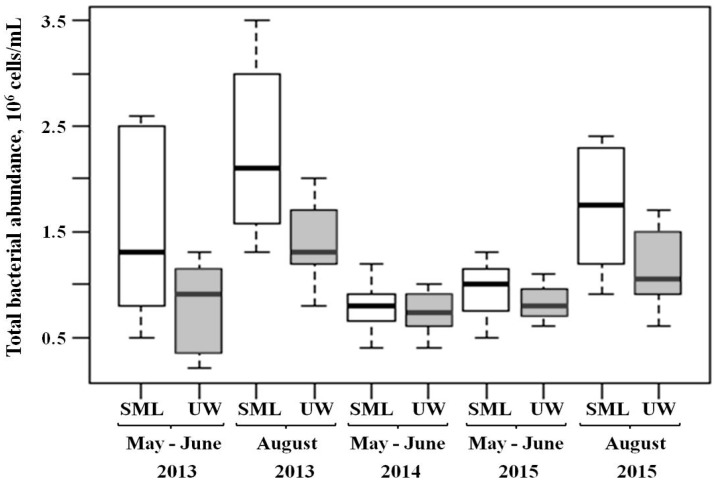
Total bacterial abundance (10^6^ cells/mL) in the SML and UW of Lake Baikal in May–June and August of 2013–2015.

**Figure 3 ijerph-15-02587-f003:**
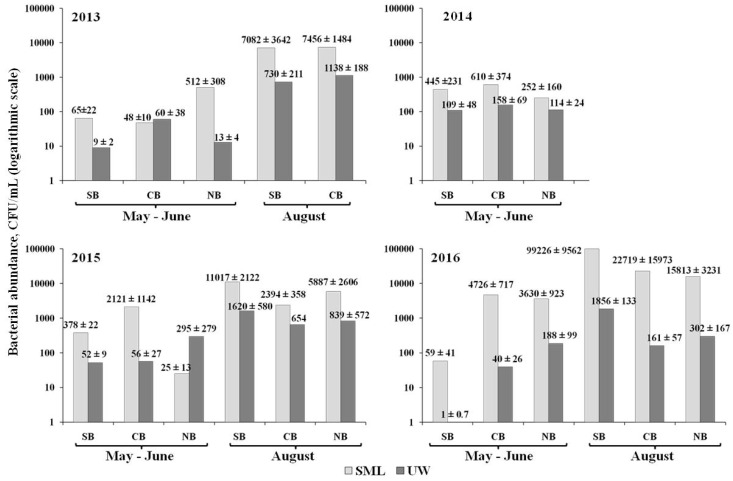
The number of cultured heterotrophic bacteria (CFU/mL) in SML and UW of southern (SB), central (CB) and northern (NB) Baikal basins in May–June and August of 2013–2016. Note. After the “±” sign, there is the standard error of the mean with a significance level of 0.05.

**Figure 4 ijerph-15-02587-f004:**
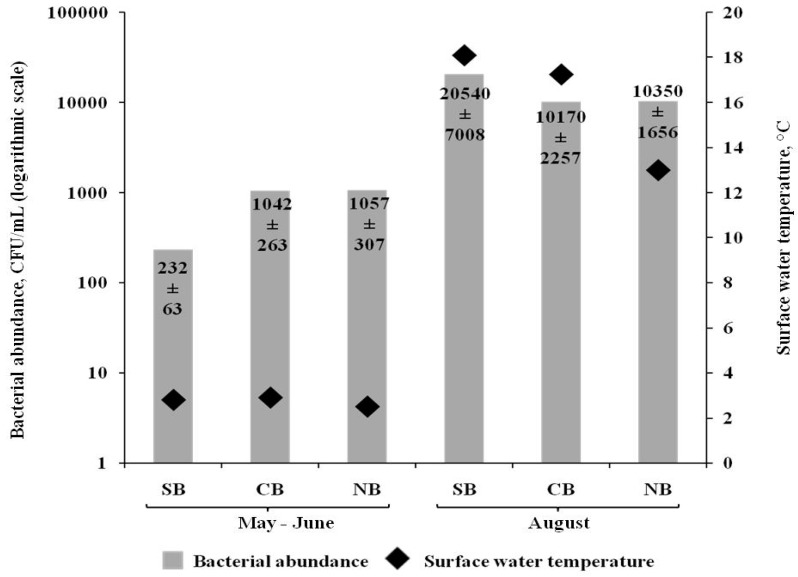
The mean values of the cultured heterotrophic bacteria number in SML of southern (SB), central (CB) and northern (NB) Baikal basins at different surface temperatures in May–June and August of 2013–2016. Note. The values of standard error of the mean at a significance level of 0.05 serve as the confidence interval.

**Table 1 ijerph-15-02587-t001:** Turbidity and chemical composition of SML and UW in Lake Baikal.

Water Layer	August of 2013			May–June of 2015		August of 2015		May–June of 2016	
	PM, mg/L	РО_4_^3−^, mg/L	NO_2_^−^, mg/L	РО_4_^3−^, mg/L	NH_4_^+^, mg/L	РО_4_^3−^, mg/L	TOC, mg C/L	РО_4_^3−^, mg/L	NO_3_^−^, mg/L
SML	13.5 ± 4.5	0.026 ± 0.006	0.010 ± 0.003	0.024 ± 0.004	0.021 ± 0.003	0.020 ± 0.005	3.5 ± 2.2	0.009 ± 0.004	0.38 ± 0.04
UW	3.6 ± 1.5	0.012 ± 0.006	0.005 ± 0.001	0.010 ± 0.002	0.007 ± 0.001	0.014 ± 0.003	1.4 ± 0.3	0.006 ± 0.004	0.36 ± 0.04

Note. PM—particulate matter, TOC—total organic carbon. The values of standard error of the mean at a significance level of 0.05 serve as the confidence interval.

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
