# Peer review of "Bacterioneuston in Lake Baikal: Abundance, Spatial and Temporal Distribution"

_ijerph, 2018, doi:10.3390/ijerph15112587_

Round 1
Reviewer 1 Report
The paper by Galachyants et al. presents data on the bacterial abundance in the SML from Lake Baikal at multiple sample points across multiple seasons. The data is novel and of interest, particularly considering a paucity of information on the bacterial community dynamics of the bacterioplankton and bacterioneuston in Lake Baikal. However there a large number of formatting issues and an expansion of the methods required before it is acceptable for publication. Details are below:
L23 PO43- needs to have subscript and superscript and for nutrients throughout the document.
L63-72 Any additional studies on the microbiology of Lake Baikal- give more background information about the biology of the system.
L75 no need to state see references
Line 78 give much greater information about the sampling site, latitude longitude co-ordinates also if you have co-ordinates for each sample site include them in the paper/supplementary information.
Line 107 A lonely sentence – include elsewhere
Line 108-110 Much more information on the exact method needed – this is the method your whole paper is built on. It is not adequate at present.
Line 111-112 Again much more information needed- how much volume was counted how many field of view don’t just give a reference write out the whole method.
Line 115-125 How was PO4 NH4 measured?
Author Response
Response to Reviewer 1 comments
Point 1: L23 PO43- needs to have subscript and superscript and for nutrients throughout the document.
Response 1: Thank you for the point. We fixed it.
Point 2: L63-72 Any additional studies on the microbiology of Lake Baikal- give more background information about the biology of the system.
Response 2: Good point. We added some more background informaiton about the biology of the system.
Point 3: L75 no need to state see references
Response 3: We changed the end of the introduction.
Point 4: Line 78 give much greater information about the sampling site, latitude longitude co-ordinates also if you have co-ordinates for each sample site include them in the paper/supplementary information.
Response 4: We added the table with all sampling stations and co-ordinates.
Point 5: Line 107 A lonely sentence – include elsewhere
Response 5: We icluded this sentence to the previous paragraph.
Point 6: Line 108-110 Much more information on the exact method needed – this is the method your whole paper is built on. It is not adequate at present.
Response 6: Good point. We added more information about the method. We are aware that this method is not adequate at present. This experiment was done to put together the number of cultured forms and total bacterial abundance, because it can give evidence of the differences in the metabolism of neuston and plancton bacteria. This suggestion was supported by the results of our experiment and by literature data.
Point 7: Line 111-112 Again much more information needed- how much volume was counted how many field of view don’t just give a reference write out the whole method.
Response 7: Good point. We described the method in details.
Point 8: Line 115-125 How was PO4 NH4 measured?
Response 8: We added this information to the text.
Reviewer 2 Report
General Overview
Authors of this manuscript have presented a study related to abundance, spatial 2 and temporal distribution of the Bacterioneuston in Lake Baikal. The authors of this paper determined For the first time, chemical composition of the surface microlayer of Lake Baikal. They found significant differences and a direct relationship between the total bacterial abundance in the surface microlayer and underlying waters of Lake Baikal. The topic is interesting, an important issue and generally well written.
The manuscript is technically correct. However, there are still some occasional grammar errors through the manuscript especially the article ‘’the’’ is missing in many places, please make a spellchecking and improve the English. Although the manuscript has a clear and state of the art Introduction, the Methodology can be repeatable of other case study and easy to implement. The reviewer recommends presenting the methodology applied in this study through a flowchart.
The results and discussion section needs further improvement, compare your findings with the other author's findings. Therefore, the reviewer recommends to further improve the manuscript before accepting it for publication. Please provide more deep discussion about your results. Please clearly state the novelty of this work.
Definitely, after the authors make respective correction and improvement, the manuscript deserves to be published in IJERPH journal.
Specific Comments
The abstract needs some improvement does not have a clear structure.
In many places’ articles ‘’the’’, ‘’a’’, ‘’an’’ is missing, please check and correct.
Please cite the following paper in line 37:
Coelho, C., Heim, B., Foerster, S., Brosinsky, A., & De Araújo, J. C. (2017). In Situ and Satellite Observation of CDOM and Chlorophyll-a Dynamics in Small Water Surface Reservoirs in the Brazilian Semiarid Region. Water, 9(12), 913.
Please cite the following paper between line 49-55:
Kuriqi, A., Kuriqi, I., & Poci, E. (2016). Simulink Programing for Dynamic Modelling of Activated Sludge Process: Aerator and Settler Tank Case. Fresen. Environ. Bull, 25(8), 2891.
Kuriqi, A. (2014). Simulink application on dynamic modeling of biological waste water treatment for aerator tank case. International Journal of Scientific & Technology Research, 3(11), 69-72.
Figure 1 has low resolution; please improve the quality.
Please consider presenting the methodology applied in this work through a flowchart; it will be much easier to understand from the readers.
Concluding Remarks
The work presented in this manuscript is an interesting topic, it needs some more efforts to improve it further. Reviewer recommend minor revision of this manuscript and publishing it only after specific improvement of the current version are made.
Author Response
Response to Reviewer 2 comments
Point 1: The abstract needs some improvement does not have a clear structure.
Response 1: The abstract was changed.
Point 2: In many places’ articles ‘’the’’, ‘’a’’, ‘’an’’ is missing, please check and correct.
Response 2: To improve our English, we used proofreading service "Proof-Reading-Service.com" (https://www.proof-reading-service.com/en/?gclid=Cj0KCQiAw5_fBRCSARIsAGodhk-9qjn0GLQIAfAnyNGnC0iDB_x0t4Qerhr59GWATeh2JXVfG8LkyWUaAv1NEALw_wcB). Managers of that service asked to contact them if there are any problems with the quality of proofreading.
Point 3: Please cite the following paper in line 37:
Coelho, C., Heim, B., Foerster, S., Brosinsky, A., & De Araújo, J. C. (2017). In Situ and Satellite Observation of CDOM and Chlorophyll-a Dynamics in Small Water Surface Reservoirs in the Brazilian Semiarid Region. Water, 9(12), 913.
Response 3: We cited the paper.
Point 4: Please cite the following paper between line 49-55:
Kuriqi, A., Kuriqi, I., & Poci, E. (2016). Simulink Programing for Dynamic Modelling of Activated Sludge Process: Aerator and Settler Tank Case. Fresen. Environ. Bull, 25(8), 2891.
Kuriqi, A. (2014). Simulink application on dynamic modeling of biological waste water treatment for aerator tank case. International Journal of Scientific & Technology Research, 3(11), 69-72.
Response 4: We cited the papers.
Point 5: Figure 1 has low resolution; please improve the quality.
Response 5: The quality of Figure 1 was improved.
Point 6: Please consider presenting the methodology applied in this work through a flowchart; it will be much easier to understand from the readers.
Response 6: We presented the methodology through a flowchart and added it to the supplementary information.